# Predictability of Marine Heatwaves: assessment based on the ECMWF seasonal forecast system

Eric de Boisseson[1], Magdalena Alonso Balmaseda[1]

[1]European Centre for Medium-range Weather Forecasts, Reading, RG2 9AX, United Kingdom

5  *Correspondence to*: Eric de Boisseson (Eric.Boisseson@ecmwf.int)

**Abstract.**

Marine heatwaves (MHWs), defined as prolonged period of extremely warm sea surface temperature (SST), have been receiving a lot of attention in the past decade as their frequency and intensity increase in a warming climate. This paper investigates the extent to which the seasonal occurrence and duration of MHWs can be predicted with the European Centre for Medium-range Weather Forecast (ECMWF) operational seasonal forecast system. The prediction of the occurrence of MHW events, the number of MHW days per season, their intensity and spatial extent is derived from seasonal SST forecasts and evaluated against an observation-based SST analysis using both deterministic and probabilistic metrics over the 1982-2021 period. Forecast scores show useful skill in predicting the occurrence of MHWs globally for the two seasons following the starting date. The skill is the highest in the El-Niño region, the Caribbean, the wider Tropics, the north-eastern Extra-tropical Pacific and Southwest of the Extra-tropical basins. The skill is not as good for other midlatitudes eastern basins, nor for the Mediterranean, the forecast system being able to represent the low frequency modulation of MHWs but showing poor skill in predicting the interannual variability of the MHW characteristics. Linear trend analysis shows an increase of MHW occurrence at a global scale, which the forecasts capture well.

## 1 Introduction

Marine heatwaves (MHWs) are defined as prolonged periods of anomalously warm sea surface temperature (SST) that can be characterized – among other - by their duration, intensity and spatial extent (Hobday et al, 2016). Due to their potential impact on marine ecosystems and the associated marine economy (Smith et al., 2021), MHW events have received a wide coverage over the past few years. High resolution operational SST analysis products covering the whole satellite period, from the early 1980s to near-real time, allow to monitor the real time evolution of such events as well as inventorying and describing events from the past four decades. Darmaraki et al [2019], Bonino et al [2022], Juza et al [2022] and Dayan et al [2023] for example looked in details at MHWs in the Mediterranean Sea, describing their duration, intensity, frequency but also long-term trends and possible future evolution. Iconic MHW events such as "the Blob" and its successor ("the Blob 2.0") in the north-eastern Extra-tropical Pacific have been described and investigated in depth in terms of attribution (Bond et al. 2015; Gentemann et al 2017; Amaya et al. 2020, de Boisseson et al., 2022) but also of impacts on the ecosystems (McCabe et al, 2016; Laurel et al, 2020; Barbeaux et al, 2020; Michaud et al, 2022).

The ability to predict MHWs in advance would allow actors of the marine industries to make decisions to limit the impact on ecosystems. For example, the return of "the Blob" in 2019 and the 2020 outlook led the US federal cod fishery in the Gulf of Alaska to close for the 2020 season as a precautionary measure as the number of cods in the area was deemed too low (Earl 2019). As a response to extreme events in the Tasman Sea (Oliver et al., 2017) and the Coral Sea (Kajtar et al, 2021), MHW forecasts on both sub-seasonal and seasonal timescales have been investigated in Australian Seas (Hobday et al, 2018; Benthuysen et al, 2021). More recently, Jacox et al (2022) investigated the predictability of MHWs on a global scale from an

ensemble of six climate models. Their results showed that forecast skill was mostly region dependent, with the eastern Equatorial Pacific region being predictable with the longest lead time. Seasonal forecasts of SST are routinely conducted by major forecasting centres mainly to predict the evolution of climate modes such as the El Nino Southern Oscillation (ENSO). Seasonal MHW forecasts can be inferred as by-product of such SST forecasts as shown by Jacox et al. (2022).

The present study follows a similar approach using the SST outputs from the ECMWF ensemble seasonal forecast system (Johnson et al, 2019) to evaluate its ability to predict MHW events on a global scale both in deterministic and probabilistic sense. A selection of regions will be investigated in more details. The main purpose of this work is to present a functional way to routinely characterise MHWs in an operational seasonal forecast system and to evaluate the forecast skill. Section 2 provides a description of the forecasting system, the verification datasets, and the methods for MHW detection and skill assessment. Section 3 presents the results regarding the spatial distribution of the skill, regional aspects, and trends. The manuscript finishes with a brief summary and outlook.

## 2 Products and methods

### 2.1 The seasonal forecasting system

The ECMWF seasonal forecast system 5 (SEAS5; Jonhson et al., 2019) is used to assess the skill in predicting MHWs over the 1982-2021 period. SEAS5 is a state-of-the-art seasonal forecast system, with a particular strength in ENSO prediction, and a member of the Copernicus Climate Change Service (C3S) multi-model seasonal forecast product. SEAS5 is based on the ECMWF Earth System model that couples atmosphere, land, wave and ocean and sea-ice. The atmospheric, land and wave components are embedded in the ECMWF Integrated Forecast System (IFS) model cycle 43r1. The atmosphere in the IFS uses a TCo319 spectral cubic octahedral grid (approximately 36-km horizontal resolution) with a 20 min time step. There are 91 levels in the vertical, with a model top in the mesosphere at 0.01 hPa or around 80 km. Initial conditions for the IFS are taken from ERA-Interim (Dee et al., 2011) prior to 2017 and ECMWF operational analyses from 2017 onwards. The physical ocean model component is based on the NEMO3.4 framework (Madec, 2008) at a ¼ degree horizontal resolution and 75 vertical levels with level spacing increasing from 1 m at the surface to 200 m in the deep ocean. Ocean initial conditions for hindcasts over the 1982–2021 period are taken from the Ocean ReAnalysis System 5 (ORAS5, Zuo et al., 2019). SEAS5 ocean forecast fields are archived at both daily and monthly frequencies. SEAS5 produces a 51-member ensemble of 7-month forecasts initialised every 1st of the month.

Here we explore the seasonal skill of SEAS5 in predicting the occurrence of MHW events on a global scale for forecasts starting on 1st February, 1st May, 1st August and 1st November. For each starting date, the forecast skill is estimated for the two following seasons corresponding to forecast range months 2-3-4 and 5-6-7 so that our study equally covers MHW happening in spring (March-April-May, MAM), summer (June-July-August, JJA), autumn (September-October-November, SON) and winter (December-January-February, DJF). The first 25 members of each forecast date are used for this assessment.

**2.2 Verification dataset**

The SST forecast from SEAS5 are evaluated against the global SST reprocessed product from the European Space Agency Climate Change Initiative (ESA-CCI) and C3S available on the Copernicus Marine Service catalogue (referred to as ESA-CCI SST in the following). ESA-CCI SST provides daily L4 SST fields at 20 cm depth on a 0.05-degree horizontal grid resolution, using satellite data from the (Advanced) Along-Track Scanning Radiometer ((A)ATSRs), the Sea and Land Surface Temperature Radiometer (SLSTR) and the Advanced Very High Resolution Radiometer (AVHRR) sensors (Merchant et al., 2019) and produced by running the Operational Sea Surface Temperature and Sea Ice Analysis (OSTIA) system (Good et al., 2020). Daily SEAS5 SST forecast fields are retrieved on a regular 1x1 degree on the Copernicus Data Store (CDS) and compared to ESA-CCI SST fields interpolated on the same regular grid.

**2.3 Marine heatwave detection**

MHW events in SST timeseries from both SEAS5 forecasts and ESA-CCI are detected over the 1982-2021 period following loosely the definition by Hobday et al (2016). For both SEAS5 SST and ESA-CCI SST, a daily timeseries of the SST $90^{th}$ percentile is computed over the common reference period of 1993-2016, the same reference period used by the C3S multi-model seasonal forecast charts (https://climate.copernicus.eu/charts/packages/c3s_seasonal/). Although the $90^{th}$ percentile threshold is estimated from the 1993-2016 climate, the MHW detection is applied for the whole 1982-2021 period. A 5-day running mean is applied to the daily ESA-CCI SST timeseries to filter out freak anomalies that would not fit the "extended period" criterion of the MHW definition. Then, we count the number of days per season where the SST exceeds the $90^{th}$ percentile over the 1982-2021 period. This is what we refer as the number of MHW days in the following. The maximum SST anomaly with respect to the 1993-2016 climatology during the MHW days is taken as the peak temperature of the MHW occurring during a given season. For SST forecasts, the detection method is similar to ESA-CCI SSTs. The daily forecast SST $90^{th}$ percentile timeseries is computed from 25 members of the SEAS5 ensemble over the 1993-2016 reference period. The number of MHW days and the maximum MHW temperature anomalies are then estimated for seasons corresponding to months 2-3-4 and 5-6-7 of the SST forecasts following the same procedure as for the ESA-CCI product. The probability of forecasting a MHW event in a given season is estimated at each grid point as the percentage of ensembles in which the number of MHW days is greater than five.

**2.4 Skill scores**

**2.4.1 Mean Square Skill Score**

To estimate the Mean Square Skill Score (MSSS), two components are needed: i) the Mean Square Error (MSE) of the MHW forecasts with respect to MHW as captured in ESA-CCI and ii) the standard deviation from the mean of a given MHW characteristic as captured in ESA-CCI. The MSSS is estimated for the forecast ensemble mean at every grid point for the period 1982-2021 as follows:

$$MSSS = 1 - \frac{MSE}{STD_o} \tag{1}$$

where,

$$MSE = \frac{1}{N}\sum_{i=1}^{N}(F_i - O_i)^2 \tag{2}$$

and,

$$STD_o = \frac{1}{N}\sum_{i=1}^{N}(O_i)^2 \tag{3}$$

where, Fi is the forecast ensemble mean anomaly for a given verification time, Oi is the corresponding verifying observation anomaly, and N is the total number of verification instances over the 1982-2021 period. MSSS is here estimated for the number of MHW days.

### 2.4.2 Multiyear trend and correlation maps and area-averaged timeseries

The long-term linear trend of the number of MHW days is computed for both SEAS5 ensemble mean and ESA-CCI. Reports of a trend toward more frequent and longer MHWs over the recent decades (Oliver et al., 2018; Collins et al., 2019) indicate a distinctive multi-year signal in observation-based SST analyses such as the ESA-CCI product. Here, the aim is to assess how well (or not) SEAS5 represents such multi-year trend. Trend errors will potentially degrade forecast scores and indicate deficiencies in either the model or the initialization. Maps of temporal correlation (with 95% significance, following DelSole and Tippett, 2016) between MHW ensemble mean forecast and observations over the 1982-2021 period are also produced for every start date and their corresponding two verifying seasons. These maps will give additional insights on the ability of the forecast to represent the multi-year signal. Area averaged timeseries of MHW characteristics are also used to evaluate the forecast system performance for individual events in regions of interest and will complete the trend and correlation diagnostics. MHW characteristics are estimated at grid points where the number of MHW days is greater than or equal to five. Such characteristics include the number of MHW days per season, the maximum amplitude during that period and the spatial extent. The spatial extent is estimated as the percentage of grid points in the considered area where the number of MHW days per season is at least five.

### 2.4.3 Relative Operator Characteristic

The relative (or receiver) operating characteristic (ROC, Swets 1973; Mason 1982; Mason and Graham 1999) is a way of assessing the skill of a forecasting system by comparing the hit (true positive) rate and the false-alarm (false negative) rate that is commonly used for weather forecasting (Stanski et al., 1989; Buizza and Palmer, 1998). The ROC is here computed at every grid point using: (i) the forecast probabilities for MHW for a given start date and verifying season inferred from the SEAS5 SST forecasts (as defined in Section 2.3) and (ii) the MHW occurrence (at least 5 MHW days) in the ESA-CCI product for the

corresponding season. Both the true and false positive rates are estimated for a comprehensive range of forecasts probabilities based on the forecast ability to capture MHW events as detected in the ESA-CCI SST fields over the 1982-2021 period. From there, ROC curves can be plotted and potentially used to select the trigger MHW probability threshold for an event that provides the best trade-off between true positive rate and false alarm rate. The ROC score is computed from the ROC curve as the normalised area under the curve (AUC, Stanski et al. 1989), where an AUC close to 0.5 indicate little to no skill while an AUC

close to one indicate high skill. In this study both the ROC curve and score are computed over a selection of regions of interest but also at every grid point to give insight into the spatial distribution of seasonal MHW forecast skill.

## 3 Results

### 3.1 Seasonal forecast skill for marine heatwaves: spatial distribution

Both correlation and MSSS of the number of MHW days per season are computed with respect to the reference dataset from
ESA CCI. These scores are deterministic in that they are inferred from the ensemble mean of the seasonal forecasts. The correlation estimates the ability of the seasonal system to reproduce the time evolution of the ESA CCI data in terms of number of MHW days. In all seasons, the highest correlations are found over the Pacific Cold Tongue where El Nino events occur and in the wider Tropics (Fig. 1). Correlations remain relatively high in the eastern Tropical Pacific as well as in the Equatorial Atlantic and Indian Oceans in the second season for SON and DJF (Fig. 1e,f), reflecting the ability of the seasonal system to
predict and persist El Nino conditions over autumn and winter. The drop in skill for JJA in the second season (Fig. 1b) in these areas is likely related to the spring predictability barrier (Webster and Yang, 1992; Balmaseda et al, 1995). High and significant correlations are seen in Extratropical areas such as the north-eastern Pacific and the Southern Ocean (particularly over the Pacific sector in MAM and JJA, Fig. 1a,c) where MHW occurrence is influenced on longer timescales by climate modes like the Pacific Decadal Oscillation (PDO), the North Pacific Subpolar Gyre Oscillation (NPGO; Di Lorenzo et al, 2008) and the
Interdecadal Pacific Oscillation (IPO) (Holbrook et al, 2019).

The MSSS indicates how close to the observed quantity the forecast gets in terms of number of MHW days. In all seasons, the highest score is again over the Pacific Cold Tongue where El Nino events occur (Fig. 2). The footprint of ENSO is partly visible in both Tropical Indian and Atlantic basins where MHW occurrence and predictability is also likely to be influenced
by climate modes such as the Indian Ocean Dipole (IOD) and the North Atlantic Oscillation (NAO), respectively (Holbrook et al, 2019). The north-eastern Extra-tropical Pacific is one of the only midlatitude region with significant MSSS values, from spring to autumn (in the first forecast season only, Fig. 2a,c,e). As expected, MSSS degrades in the second season of the forecast and most of the skill is concentrated over the Pacific cold tongue in SON and DJF (Fig. 2d,f), strongly suggesting links between MHW and ENSO predictability. Overall, MSSS and correlation values larger than zero are widespread and
mostly significant (especially correlations), indicating that, even at these long lead times, the seasonal forecasts are more skilful than climatology.

The ROC allows to evaluate the seasonal forecasts in terms of ability to detect the presence of a MHW event within a season. Such score can help decision-making to prepare for or mitigate the impact of a likely MHW event when the forecast probability exceeds a certain threshold. Maps of AUC provide indications of the area where there is MHW forecast skill. For forecast range 2-to-4 months (season one), values of AUC over 0.5 are found almost everywhere (Fig. 3a,c,e,g). The largest values are found in both the Nino 3.4 and 4 regions, reflecting once more the ability of SEAS5 to predict and persist El Nino conditions. Overall, AUC is high over the Tropics and Sub-Tropics in all basins. The north-eastern Extra-tropical Pacific, where "the Blob" happened, shows high skill in all seasons. Skilful MHW prediction are seen in the western Tropical Atlantic mainly for MAM and JJA (Fig. 3a,c), the Tropical Indian for MAM, SON and DJF (Fig. 3a,e,g) and over the Maritime Continent mainly for JJA (Fig. 3c). The skill overall decreases in the forecast range 5-7 months (season 2, Fig. 3b,d,f,h), with the highest values of AUC in both Tropical Pacific and Indian Oceans, the north-eastern Extra-tropical Pacific and the Pacific sector of the southern Extra-tropics. The ROC score complements and confirms the results from both MSSS and correlation. The ROC maps indicate the areas where the forecast system can predict observed MHW events on seasonal timescales. MSSS and correlation show the accuracy of such predictions in terms of length and interannual variability of extreme SST events. This set of skill indicates that even at long lead times the seasonal forecasts from SEAS5 show useful skill in predicting the occurrence of MHW events.

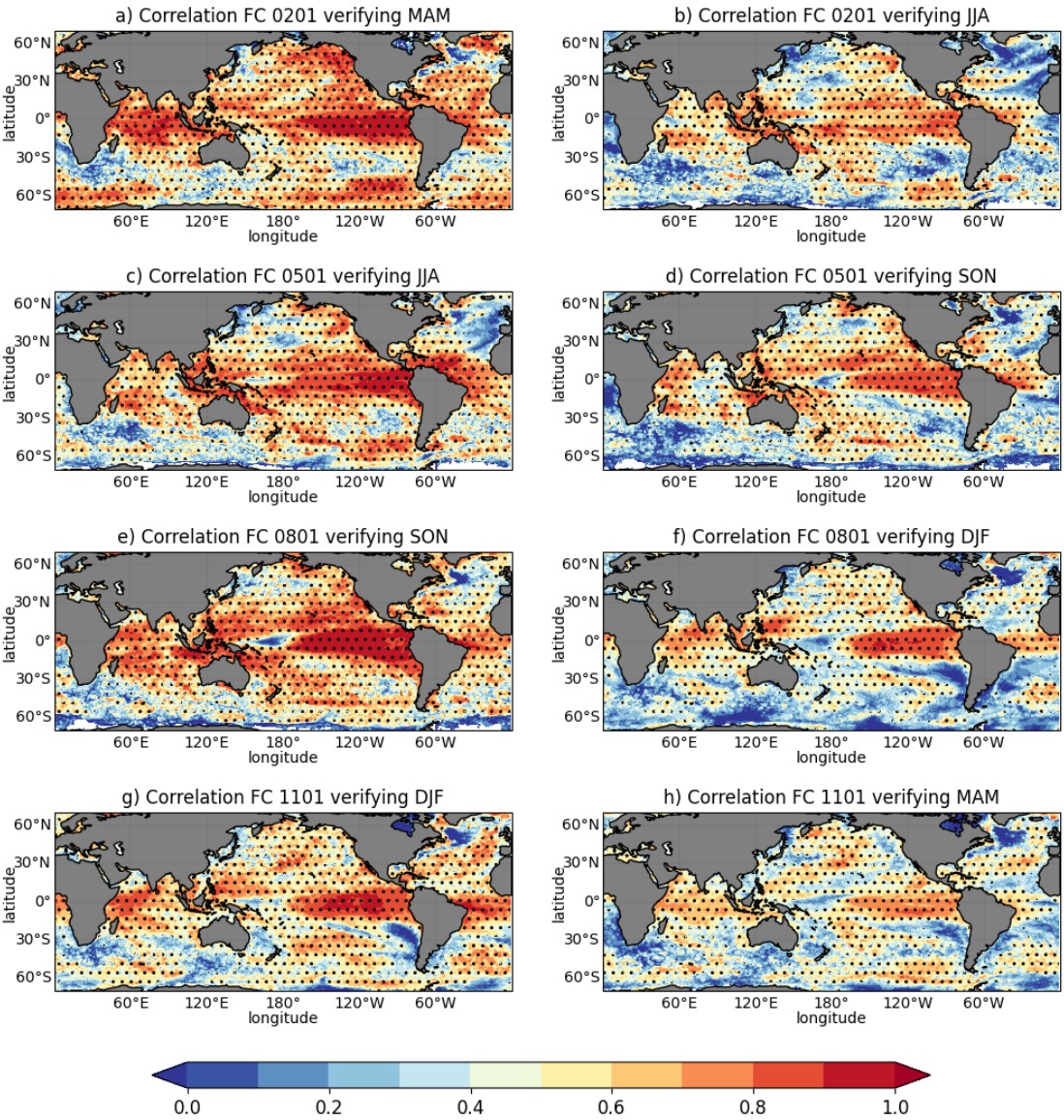

**Figure 1 Maps of interannual correlation between the number of MHW days forecasted in season one (months 2-3-4) and two (months 2-3-4) and the observed number of MHW days for starting dates on: the 1st February verifying (a) MAM (March-April-May) and (b) JJA (June-July-August), the 1st May verifying (c) JJA and (d) SON (September-October-November), the 1st August verifying (e) SON and (f) DJF (December-January-February), and the 1st November verifying (g) DJF and (h) MAM. Forecasts for the period 1982-2021 are verified against ESA-CCI SST product. The hatches indicate area in which the scores are significant. Significance for both MSSS and correlation is estimated following DelSole and Tippett (2016).**


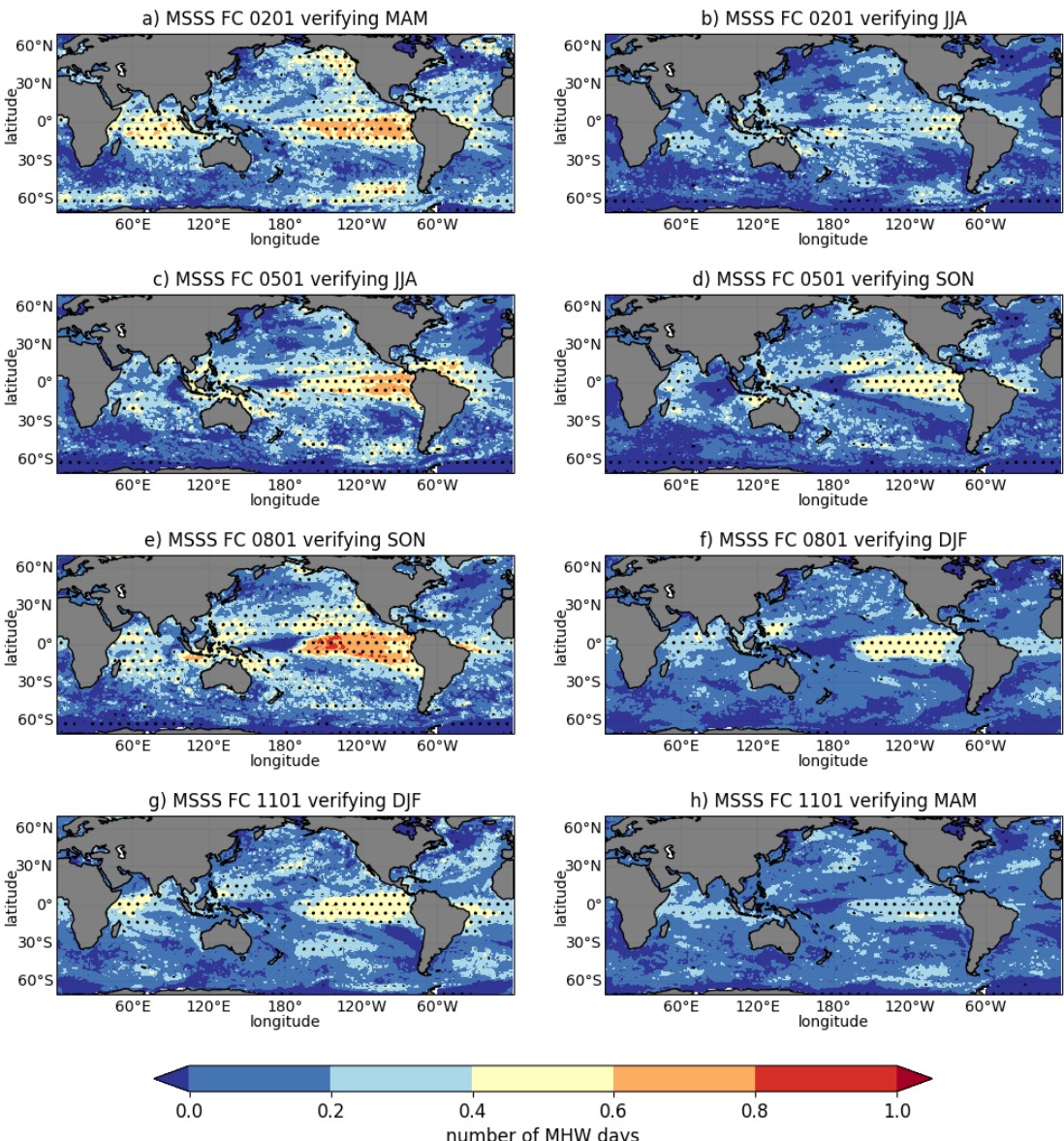


**Figure 2 Maps of mean square skill score of the number of MHW days for season one (months 2-3-4) and two (months 5-6-7) of the forecast starting on: the 1st February verifying (a) MAM (March-April-May) and (b) JJA (June-July-August), the 1st May verifying (c) JJA and (d) SON (September-October-November), the 1st August verifying (e) SON and (f) DJF (December-January-February), and the 1st November verifying (g) DJF and (h) MAM. Forecasts for the period 1982-2021 are verified against ESA-CCI SST product.**
**The hatches indicate area in which the scores are significant. Significance for both MSSS and correlation is estimated following DelSole and Tippett (2016).**

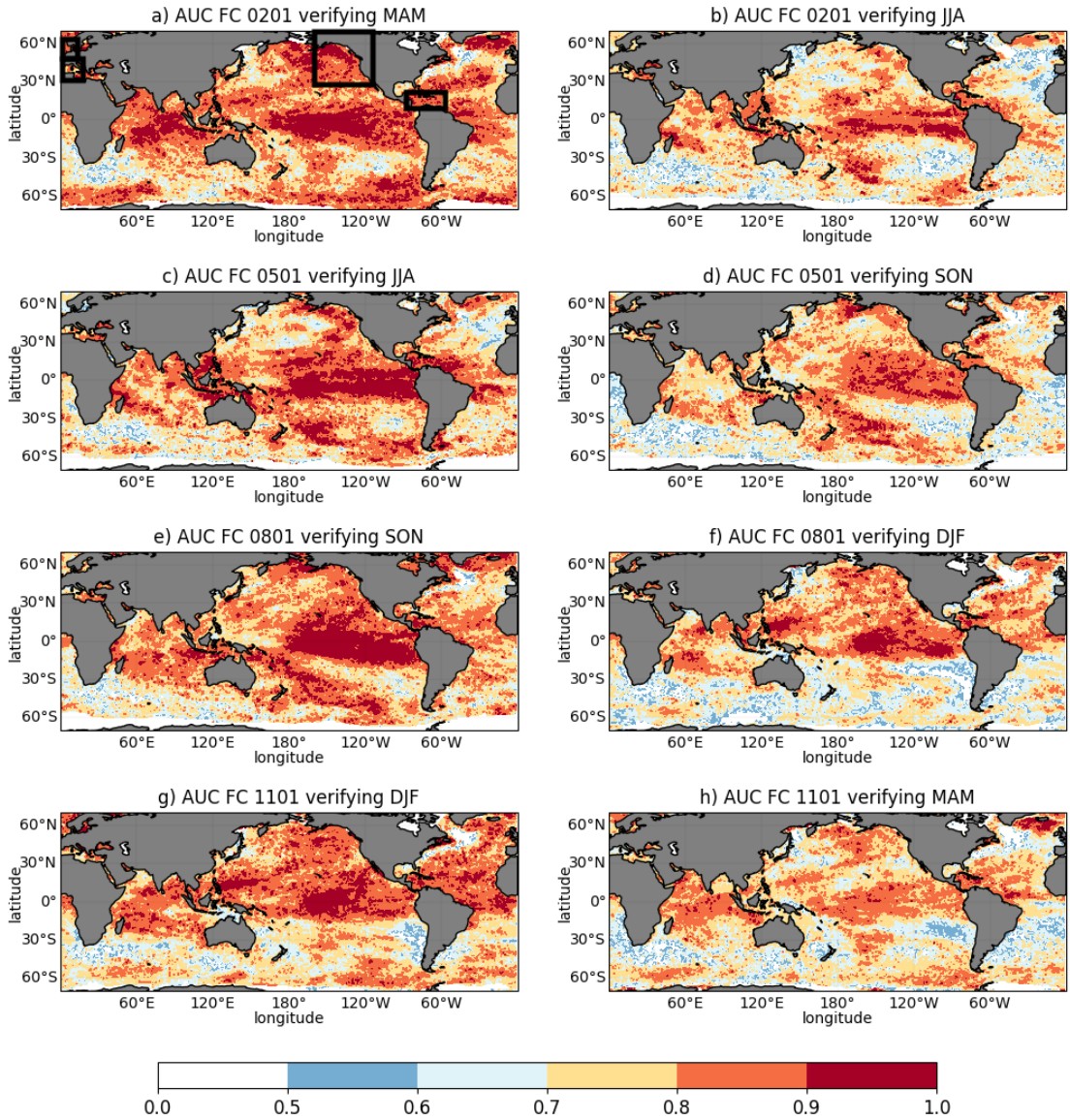

**Figure 3 Maps of the Area Under the Curve (AUC) forecasted in season one (months 2-3-4) and two (months 2-3-4) for starting dates on: the 1st February verifying (a) MAM and (b) JJA, the 1st May verifying (c) JJA and (d) SON, the 1st August verifying (e) SON and (f) DJF, and the 1st November verifying (g) DJF and (h) MAM. The AUC is derived from the ROC curves estimated from the probability of predicting at least 5 days of SST in the 90th percentile during a season. The boxes on panel (a) indicates the 4 areas (north-eastern Extra-tropical Pacific, Caribbean, West Mediterranean, and North Sea) used to produce Figs. 4, 5 and 6.**

## 3.2 Seasonal forecast skill for marine heatwaves: regional aspects

Looking at areas outside of the Nino region brings more nuance. The ROC is estimated for a selection of regions where MHWs could impact marine sectors such as fisheries or aquaculture. Figure 4 shows the ROC curve for seasonal forecasts starting on 1st February and 1st May and verified for JJA. The ROC curve shows very high skill in the north-eastern Extra-tropical Pacific (Figure 4a) and even higher skill in the Caribbean (Fig. 4b) for JJA. There is however a substantial reduction of the AUC in JJA for the February forecast. The skill is much lower in the West Mediterranean and rather poor in the North Sea whatever the forecast range (Fig. 4c,d). This disparity in skill reflects the known difference of performance of seasonal forecasting systems between the Tropics and Extra-tropics (especially over Europe).

Timeseries of MHW characteristics for these areas complement the ROC curves showing to which extent specific MHW events are captured by the seasonal forecasts. Figures 5 and 6 show the number of MHW days, the maximum amplitude and the spatial extent (in terms of proportion of the area affected by a MHW) in JJA over the period 1982-2021 in the February and May forecasts and the ESA-CCI product. In the north-eastern Extra-tropical Pacific (Fig. 5a,c,e), the seasonal forecast can capture the major JJA events of 1997, 2004, 2013-2015 (aka the "Blob") and 2019, although the severity of the events was underestimated in 2004. The range of maximum amplitude of the events is mostly similar to observations from 1982 to 2010 and then slightly underestimated from 2010-onwards. The time evolution of the spatial extent of MHWs is well captured (albeit the large spread), suggesting the seasonal forecast system can represent the correct spatial patterns. Both forecast starting dates show similar ability in predicting JJA MHW characteristics. The thermal memory of the ocean has been shown to impact the predictability of MHW and improved seasonal skill in the north-eastern Pacific from 2017 has been linked to an increase in the ocean stratification preconditioning the ocean to the occurrence of extremely warm events at the surface (de Boisseson et al, 2022). The state of the north-eastern Extra-tropical Pacific Ocean is influenced on synoptic to seasonal timescale by local variations in atmospheric conditions (Holbrook et al., 2019) that show relatively low predictability in SEAS5 (Johnson et al, 2019), hence impacting the accuracy of the MHW forecast. Jacox et al [2022] showed that the skill of seasonal MHW prediction in the north-eastern Pacific (close to the North American coast) is relatively improved when ENSO is an active state with respect to a neutral state. This link to ENSO could partly explain the better performances in 1997 and 2015 (strong El-Nino years) with respect to 2004 (a moderate to neutral ENSO year) for example. Aside from these modes of interannual variability, the timeseries, the number of MHW days and spatial extent appear dominated by low frequency variability or trends, which will influence the predictability. We will return to this point later in the next section.

In the Caribbean (Fig. 5b,d,f), the prediction of both the number of MHW days and the spatial extent is quite accurate especially for JJA 1998, 2005 and 2010 in the May forecast. This forecast looks confident with relatively low spread. The amplitude of the events is relatively low in both the forecasts and the observations. The forecasts are however not performing well in 1995, 2011, 2017 and 2020 for events that cover most of the region. The February forecast is less skilful in capturing the length of

the 1998, 2005 and 2010 MHW events. Cetina-Heredia and Allende-Arandía [2023] linked the development of MHW in the Caribbean in 1998 and 2010 to predictable El-Nino conditions. MHW in the Caribbean are also heavily influenced by the seasonal fluctuations of the Intertropical Convergence Zone (ITCZ) that usually come with weaker surface winds and weaker heat loss from the ocean to the atmosphere over the boreal summer (Fordyce et al., 2019). The well-predicted 2005 MHW event coincides with atmospheric conditions including particularly weak easterlies and anomalous shortwave radiation (Foltz and McPhaden, 2006) that started in winter and persisted over the summer. MHW occurrence in the Caribbean have also been linked to modes of variability such as the NAO (Holbrook et al, 2019) and the East-Atlantic Pattern (EAP) that are less predictable (Dunstone et al, 2023) and could affect MHW forecast performances.

In both the West Mediterranean and the North Sea (Fig. 6), the performance is not as good for both starting dates. Although the forecast system tends to capture the low frequency modulation of MHW (trend in the West Mediterranean and decadal modulation in the North Sea), especially in term of spatial extent (Fig. 6e,f), it does not appear skilful in predicting the interannual variability, producing false alarms and missing major events such as the one following the 2003 European heatwave. The low performance in the West Mediterranean agrees with Jacox et al. [2012] that show consistently low forecast probabilities for MHW in the area over the 1991-2020 period. McAdam et al [2023] also show poor forecast skill in the Mediterranean (albeit in the Eastern basins) at the ocean surface but argue that predictability can be found at the subsurface. The low skill in the North Sea is also in agreement with these two publications. There is little surprise in such lack of skill given the well-documented difficulties of SEAS5 in these regions (Calì Quaglia et al, 2022) that poorly predicts both NAO and SSTs in the north-western Atlantic (Johnson et al, 2019) and shows little skill in capturing some major atmospheric heatwave events that would impact the ocean surface (Prodhomme et al, 2022).

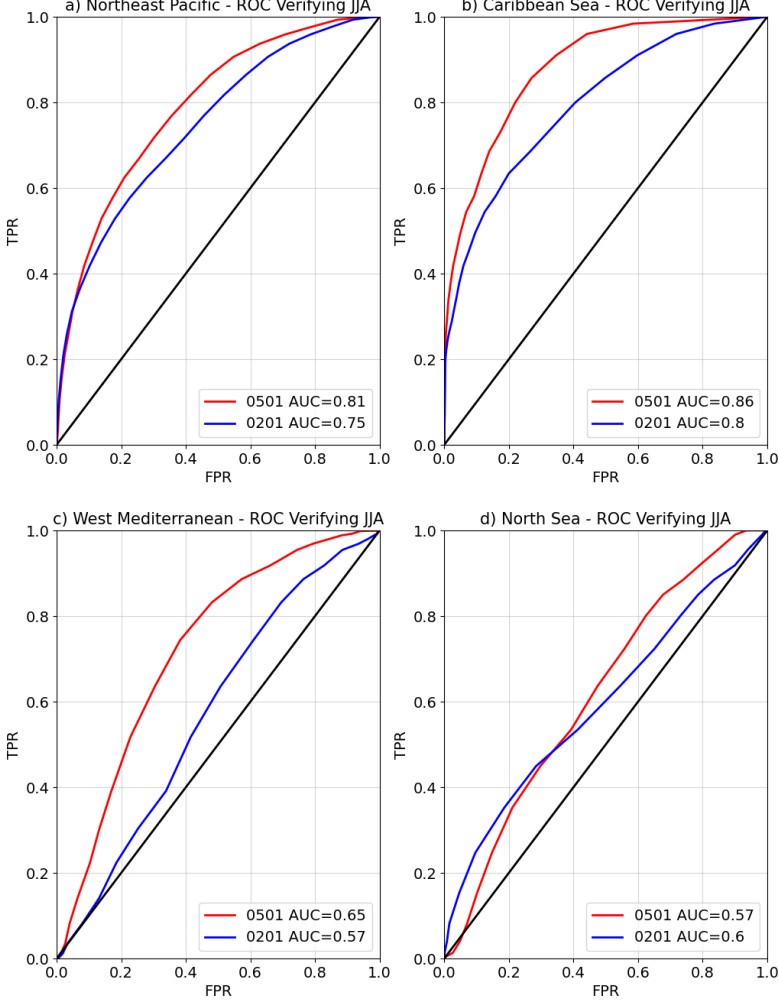

**Figure 4 ROC curve for the JJA MHW forecast starting on 1st February (blue) and 1st May (red) in a) the north-eastern Extra-tropical Pacific, b) the Caribbean, c) the West Mediterranean and d) the North Sea. The areas are defined on Figure 3a.**

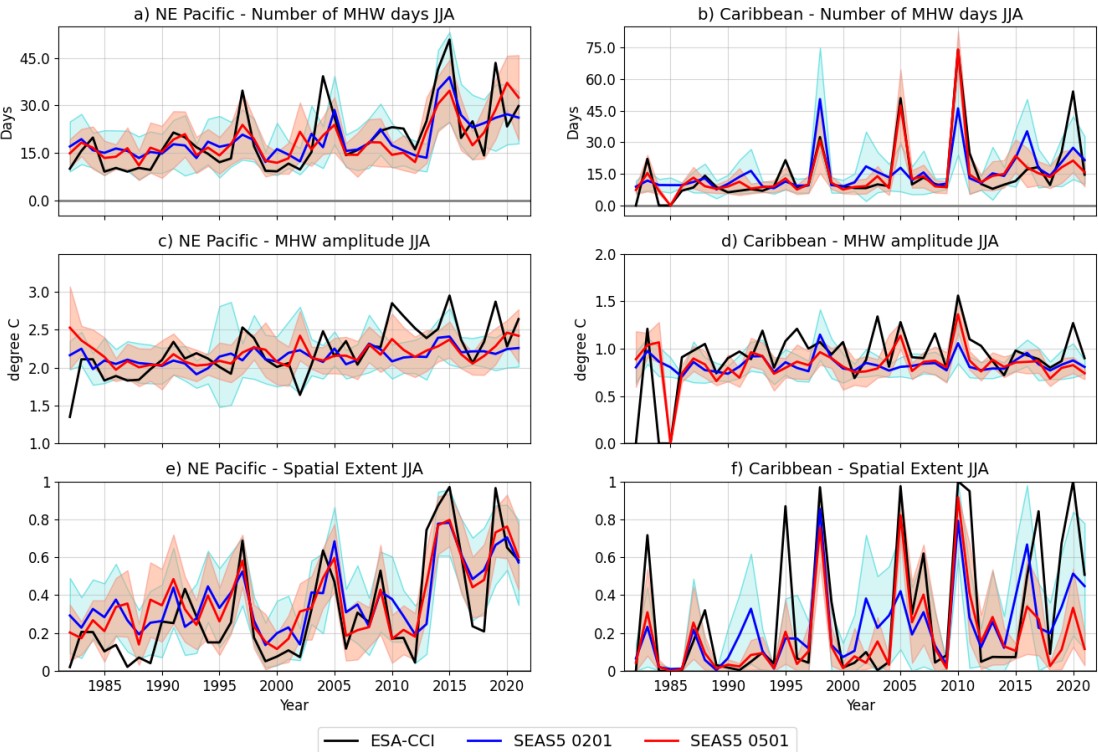

**Figure 5 Timeseries of MHW characteristics for JJA 1982-2021 in both forecasts and observations in both the north-eastern Extra-tropical Pacific (a,c,e) and the Caribbean (b,d,f): (a,b) number of MHW days, (c,d) maximum amplitude of the MHW and (e,f) spatial extent expressed as the proportion of the full area seeing a MHW event during the season. The seasonal forecasts starting on 1st February and 1st May are in blue and red, respectively, with the solid line representing the ensemble mean and the shaded area the ensemble spread. The MHW characteristics as in the ESA-CCI product are in black.**

260

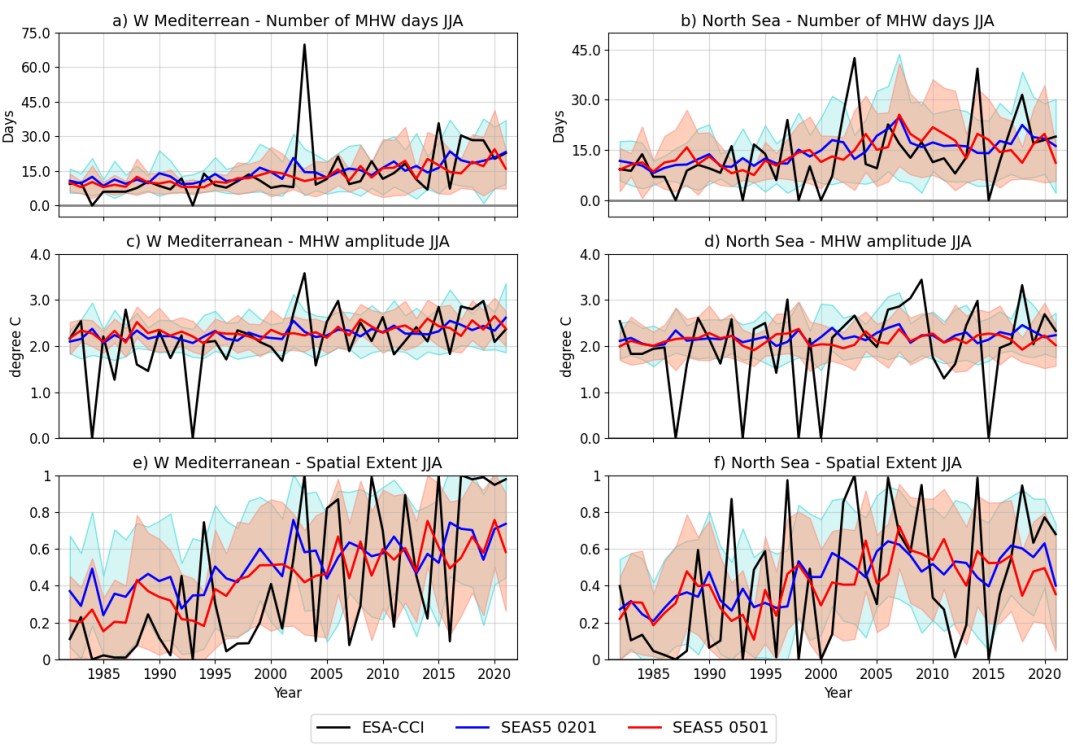

**Figure 6 Same as Figure 5 for the West Mediterranean (a,c,e) and the North Sea (b,d,f).**

## 3.2 Observed and predicted trends for marine heatwaves

The number of MHW days has been increasing since the first decades of the 20[th] century (Oliver et al, 2018), and is expected to increase further in the context of global warming (Oliver et al, 2019). Global warming has already been identified as a factor contributing to MHW occurrence leading to severe coral bleaching in the Caribbean (Donner et al, 2007). The trend in MHW days in the seasonal forecast is evaluated against observations as another assessment metric for the forecast system. Figure 7a,b displays the trends in JJA for both the ensemble mean forecast starting on 1[st] May and the ESA-CCI product over the 1982-2021 period. The number of MHW days in the ESA-CCI product increases in most ocean regions, the Pacific cold tongue and parts of the Southern Ocean being the exceptions. The forecast is able to capture most of the observed features, with hot spots over the Pacific warm pool, in the Tropical Indian Ocean and in the Southwest Pacific off New Zealand. The forecasted trends are however often weaker than the observed ones, especially in the Tropics, the north-eastern Extra-tropical Pacific and the north-western Subtropical Atlantic. Conclusions are similar for trends in MAM, SON and DJF for forecasts starting on 1[st] February, 1[st] August and 1[st] November (not shown).

Figure 7c,d displays the trends in mean SST in JJA for both forecast and observation. The forecast trends mostly capture the observed ones in the Tropics but are underestimated (overestimated) in the northern (southern) Extra-tropics. Both forecast and observations show different spatial patterns on the trends of seasonal means of SST and number of MHW days. In the Tropical Indian Ocean, northern Subtropical Eastern Pacific and Caribbean/north-western Subtropical Atlantic, the trends in

number of MHW days appear more intense than the trends in seasonal mean SST. The colder high-latitude regions bordering the Arctic, by contrast, show more pronounced trends in seasonal SST means than in number of MHW days. These results illustrate the non-linear nature of the climate change (e.g. in that over warm convective areas it is difficult to increase the mean SST, while still possible to increase the occurrence of MHW events) and highlights the importance of dedicated diagnostics to detect changes in extremes.

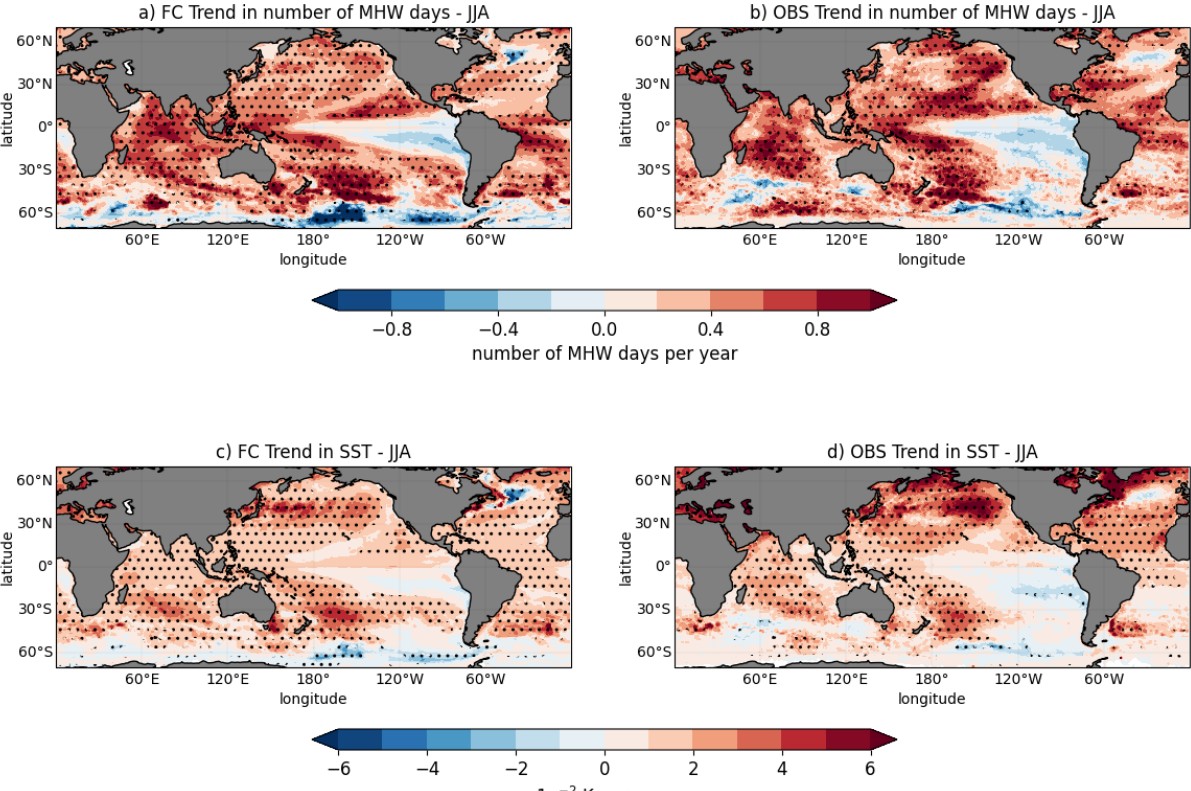

**Figure 7 Maps of the trend in number of MHW days (in number of days per year) over the 1982-2021 period in JJA for the ensemble mean seasonal forecast and the ESA-CCI SST analysis, respectively; c,d) Maps of the trend in mean SST (in K per year) over the 1982-2021 period in JJA for the ensemble mean seasonal forecast and the ESA-CCI SST analysis, respectively. The hatches indicate area in which the trends are significant. Significance is estimated following DelSole and Tippett (2016)**

## 4 Discussion and conclusions

Global daily seasonal SST forecasts are or can be routinely output by operational forecasting centres. Predicted MHW characteristics can be derived from such forecasts and could eventually be delivered to stakeholders from the marine economy and management communities. This study evaluates the skill of the ECMWF SEAS5 system in predicting the occurrence of MHWs on seasonal timescales. This work comes after a series of recent publications on seasonal MHW predictions (Spillman

et al, 2022; Jacox et al, 2022; McAdam et al, 2023) that are based on different seasonal prediction systems. In these studies, methods are different, with Jacox et al. [2022] using monthly forecast timeseries while McAdam et al [2023] are focusing more on forecasts of the ocean subsurface. Both Spillman et al [2021] and Jacox et al. [2022] are also investigating the predictability of more sophisticated aspects such as the onset of MHW events. In all these studies, the MHW detection is based on the widely accepted definition from Hobday et al [2016]. Here, we proposed a slightly simpler definition to make it easily applicable to a wide range of forecasting systems and allow flexibility according to the use one wants to make of a seasonal MHW forecast. In forecasts from the SEAS5 system, we counted the number of days per season in which the SST is in the 90[th] percentile. Focusing on a specific area, this method can provide seasonal forecast of the number of MHW days, the maximum amplitude of the MHWs over a season and the proportion of the area affected by MHWs. Skill evaluation in this study is mostly based on the number of MHW days. Both deterministic (MSSS, correlation and trend) and probabilistic (ROC) methods complement each other assessing different aspect of the forecast skill.

Results presented here suggest that, in the current state of the SEAS5 system, MHW prediction skill is very much area dependent, confirming conclusions from previous studies (Spillman et al, 2021; Jacox et al., 2022). The largest skill is found in the Tropics with a clear footprint of El Nino in the Eastern Pacific, highly predictable at interannual time scales (Fig. 1 and Fig. 3) for both season 1 and 2 of the forecast and consistent with the predictability of ocean and atmospheric conditions linked to ENSO (L'Heureux et al, 2020). The signature of the PDO is apparent over the north-eastern Pacific, with high predictability skill in the first season consistent with both Jacox et al [2022] and McAdam et al [2023]. This is consistent with processes highly conditioned by the ocean mixed layer but affected by the more unpredictable variability of local atmospheric circulation (Gasparin et al, 2020; de Boisseson et al, 2022). MHW occurrence in warm pool areas such as Western Pacific, the Indian Ocean and the Caribbean (Figs 4b and 5b,d,f) is well predicted by SEAS5. These areas are affected by long term trends (Bai et al, 2022; Donner et al, 2007) that slowly and consistently warm and deepen the warm pool and favour the onset of MHW. Climate modes such as the IOD and ENSO also impact the predictability of MHW in such regions, with location-dependent skill (Mayer et al, 2023). The MHWs in the North Atlantic and the northern European seas are influenced by the NAO and the Arctic Oscillation (Holbrook et al, 2019; She et al, 2020) that have limited and fast-decaying seasonal dependent skill (Scaife et al, 2014; Dunstone et al, 2023). The low skill in capturing major events in the Mediterranean showed in this study agrees with both Jacox et al [2022] and McAdam et al [2023] and is probably due to the impact of unresolved atmospheric variability (Ardilouze et al, 2017; Patterson et al, 2022). This is an area that would require further investigation with higher resolution models. That said, the low frequency modulation of MHW characteristics is captured and some level of skill in detecting the occurrence of MHW is found even at long lead times (Fig. 3 and Fig. 6).

Biases, limited representation of teleconnections and climate modes, atmospheric noise and model resolution all limit the predictability of MHW, in particular in the northern Extra-tropics. With record global atmospheric temperatures being reached in both 2022 and 2023, the current El Nino expected to lead to another hot year and recent intense and long-lasting

MHW events already reported in various basins (Marullo et al, 2023; Oh et al, 2023; Berthou et al, 2023), accurate seasonal predictions could rapidly become very valuable for decision-making to alleviate the socio-economic impacts of such extreme events (Smith et al., 2021). Extracting more MHW prediction skill from seasonal predictions could be achieved using a multi-model ensemble (Jacox et al, 2022). The MHW forecast produced for SEAS5 could be, for example, generalised to the multi-model ensemble from the Copernicus Climate Change service (C3S) and seasonal predictions of MHW parameters be a product released on a regular basis to be used as additional information by potential stakeholders. Given the nature of this study, the detection method is very general, and more prediction skill could be found devising targeted MHW indicators and thresholds according to a specific location, activity or ecosystem. While MHW events are mostly detected at the surface, impacts on ecosystems and populations happen in the subsurface. Seasonal forecast of ocean variables other than SST has so far received little attention, but recent work hints that forecast skill for the ocean heat content in the upper 300 m is comparable to the skill for SST in the Tropics, and even exceeds it in the Extra-tropics (McAdam et al. 2022). The recent study by McAdam et al (2023) actually showed that forecasting skill for MHW can be found in the 0-40m layer depending on the region of interest and the type of MHW event. Further analysing seasonal forecast of relevant ocean variables might be another avenue in providing useful skill for predicting extreme marine events such as MHW.

## Datasets

This study used the following European Union (E.U.) Copernicus service datasets:

ESA SST CCI and C3S reprocessed sea surface temperature analyses. E.U. Copernicus Marine Service Information (CMEMS). Marine Data Store (MDS). DOI: 10.48670/moi-00169 (Accessed on 14-03-2023)

Copernicus Climate Change Service, Climate Data Store, (2018): Seasonal forecast daily and subdaily data on single levels. Copernicus Climate Change Service (C3S) Climate Data Store (CDS). DOI: 10.24381/cds.181d637e (Accessed on 22-03-2023)

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
