# Peer review of "Predictability of Marine Heatwaves: assessment based on the ECMWF seasonal forecast system"

_EGUsphere, 2023_

## Author Response (AR1)

**Point-by-point response to the reviews:**

We would like to thank both the editor and the reviewers for their constructive comments. The manuscript has been modified accordingly. Please see below the point-by-point response to the reviews and a list of all relevant changes made in the manuscript.

**Reply to RC1**

>> In this manuscript, the authors evaluate the ability of ECMWF seasonal forecast system to predict marine heatwave in the two following seasons over the global ocean during the period 1982-2021. Skill scores have been defined to evaluate both spatial and temporal capability of the forecasts. The authors highlight the good skills in some regions, in particular the El-Niño region, and poor skills in another sub-basins such as the western Mediterranean and North Seas. The manuscript is well-written, the computational work is probably very important to provide such data and reach such results. Such work is valuable for the community and more widely for the society due to the societal and economic implications. Results are promising. I suggest some revisions before publishing this manuscript. In particular, I would suggest to better address the ability of the forecast to predict MHW through the various spatio-temporal scales of both atmospheric and oceanic processes involved. I would also suggest to provide more explanation and discussion about the results, as well as ideas to improve the forecast system. Please find below general and specific comments.

We would like to thank you for taking the time to review our manuscript. We took into account your constructive comments and modified the manuscript accordingly. Both the section describing prediction skill of Marine heatwaves and the discussion have been developed including references to relevant atmospheric and oceanic processes influencing their occurrence. Comparisons to recent publications on Marine heatwave forecast as well as a discussion on potential future improvements have been added to the manuscript. See below the responses to the individual comments.

**General and specific comments**

**Writing**

>> … Sea Surface Temperature (SST) --> sea surface temperature (SST)

The manuscript was corrected accordingly (L.8).

>> Marine Heatwaves --> Marine heatwaves

The manuscript was corrected accordingly.

>> Northeast Pacific --> north-eastern Pacific?

The mention "Northeast" has been changed to "north-eastern" throughout the manuscript.

>> Avoid the acronyms in the titles of (sub)sections

Acronyms have been removed from the titles of (sub)sections.

>> When citing figures, remove the description such as "left panel", "compare left and right" --> only " Fig. X"

The manuscript was corrected accordingly.

**Period of study**

>> Please check the periods in the text: 1982-2021, 1982-2020, 1993-2021. For consistency, you should have a unique period of study using a unique baseline period.

The period for evaluation is 1982-2021. Discrepancies have been corrected throughout the manuscript. Note however that there are two distinct temporal periods: the reference climatological period to define the 90th percentiles (1993-2016), and the verification period (1982-2021). The 1993-2016 reference period has been chosen to match that used to issue the multi-model seasonal forecast anomalies in the Copernicus Climate Change Service. The longer verification period allows more robust skill detection. See Section 2.3 of the manuscript.

**MHW seasonal forecasts**

>> Could you cross your results with those from recently published by McAdam et al., 2023?

>> *McAdam, R., Masina, S., & Gualdi, S. (2023). Seasonal forecasting of subsurface marine heatwaves. Communications Earth & Environment, 4(1), 225.*

Results are now compared with recent publications such as McAdam et al (2023) but also Jacox et al (2022) in both Sections 3 and 4 of the manuscript.

**Different drivers, different scales**

>> MHW can be caused by a large panel of anomalous atmospheric conditions. In the manuscript, the authors mainly focus on El-Nino. Among the climate modes, there are other climate modes involved in the generation of MHW? In addition, another atmospheric processes should be mentioned. The ability of seasonal weather forecast in predicting large-scales modes may explain the focus on El-Nino but the difficulty in predict other atmospheric processes has to be mentioned since not-well predicted weather forecasts will prevent skilful ocean forecasts.

Other climate modes indeed impact the occurrence of Marine heatwaves as shown in Holbrook et al. (2019). We now discuss the potential impacts of the Pacific Decadal Oscillation in the north-eastern Pacific, the North Atlantic Oscillation over the northern European Seas and the Mediterranean for example. Local and short-lived processes and their difficulty to be predicted are also referred to. See Sections 3 and 4 of the manuscript.

>> I recommend the following publications:

>> *Holbrook, N. J., Scannell, H. A., Sen Gupta, A., Benthuysen, J. A., Feng, M., Oliver, E. C., ... & Wernberg, T. (2019). A global assessment of marine heatwaves and their drivers. Nature communications, 10(1), 2624.*

This publication has been added as a reference in the manuscript (L.150, 155, 222 and 240).

**Introduction**

>> - Concerning the socio-economic impact, I recommend this nice article:

>> *Smith, K. E., Burrows, M. T., Hobday, A. J., Sen Gupta, A., Moore, P. J., Thomsen, M., ... & Smale, D. A. (2021). Socioeconomic impacts of marine heatwaves: Global issues and opportunities. Science, 374(6566), eabj3593.*

Thank you for the suggestion. Very interesting article indeed. This reference has been added to the manuscript (L.23 and 333).

>> - Listing the articles in the Mediterranean Sea, you could cite:

>> *Juza, M., Fernández-Mora, À., & Tintoré, J. (2022). Sub-Regional marine heat waves in the Mediterranean Sea from observations: Long-term surface changes, Sub-surface and coastal responses. Frontiers in Marine Science, 9, 785771.*

This reference has been added to the manuscript (L.26).

>> - Regarding the climate modes, you only refer to El-Nino. Another climate modes could generate MHW in others regions? Please comment and complete your manuscript.

Thank you for your comment. As said above, other climate modes indeed impact the occurrence of Marine heatwaves as shown in Holbrook et al. (2019). The manuscript now refers to modes such as the Pacific Decadal Oscillation, the North Atlantic Oscillation or climate change and refers to relevant publications. See Sections 3 and 4 of the manuscript.

**Product and methods**

>> Section 2.1. Could you precise the temporal and spatial resolution of the atmospheric model system? Are the ocean model outputs daily? Please precise.

Yes, the ocean outputs are daily. The atmosphere in the IFS uses a TCo319 spectral cubic octahedral grid (approximately 36km horizontal resolution) with a 20 min time step. There are 91 levels in the vertical, with a model top in the mesosphere at 0.01 hPa or around 80 km. This has been added to Section 2.1 of the manuscript.

>> Section 2.2. Not use the CMEMS acronyms. To be used: "Copernicus Marine Service"

We replaced "CMEMS" by "Copernicus Marine Service" as suggested (L.73).

>> Please provide references and DOI of the products used. You can check in the Copernicus Marine Service the section "How to cite".

Both references and DOI for both ESA-CCI SST and SEAS5 SST forecast fields are now provided as follows at L.345-351:

*ESA SST CCI and C3S reprocessed sea surface temperature analyses. E.U. Copernicus Marine Service Information (CMEMS). Marine Data Store (MDS). DOI: 10.48670/moi-00169 (Accessed on 14-03-2023)*

*Copernicus Climate Change Service, Climate Data Store, (2018): Seasonal forecast daily and subdaily data on single levels. Copernicus Climate Change Service (C3S) Climate Data Store (CDS). DOI: 10.24381/cds.181d637e (Accessed on 22-03-2023)*

>> Section 2.3. Could you explain why restrict the reference period to 1993-2016?

We restricted the reference period to 1993-2016 for consistency with the online C3S seasonal forecast charts that display the C3S multi-model seasonal forecast fields and verification scores (https://climate.copernicus.eu/charts/packages/c3s_seasonal/). This reference period is also commonly used for Copernicus Marine Service products as it covers the satellite altimetry observing period. It is also the period used for the recent study by McAdam et al (2023) on Marine heatwave seasonal forecasts in the north-eastern Pacific and the Mediterranean. This choice is now justified in Section 2.3 of the manuscript.

>> Section 2.4.3 Could you provide references for the ROC AUC methodology? Is the "R" relative?

"R" is for "relative" or "receiver" depending on the sources. Weather-related literature mostly uses "relative" so we sticked to it. We added a couple of sentences introducing the ROC with references to relevant literature at the beginning of section 2.4.3 of the manuscript and reference to Stanski et al. (1989) at for the use of the area under the curve diagnostic. See L.125-136.

*Stanski, H. R., L. J. Wilson, and W. R. Burrows, 1989: Survey of common verification methods in meteorology. WMO World Weather Watch Tech. Rep. 8, WMO TD 358, 114 pp.*

**Results**

>> Section 3.2 I would like to have more explanations such as line 188-189, why? L191-195, do the mentioned years correspond to strong El-Nino events (such as 1998)? Which phenomena occur these years? Which were the drivers of MHW? The ability to predict MHW is related to the atmospheric/weather forecast capability at specific temporal and spatial scales? Could discuss the scales and limitations?

More explanations have been added to Section 3.2 based on relevant publications. Some MHW events in the Caribbean are indeed linked to El-Nino events (in 1998, 2010 for example) while others (in 2005 for example) are linked to more local and shorter timescales atmospheric and oceanic processes. A discussion on the limits of seasonal MHW prediction has been added in both Sections 3 and 4 of the manuscript. Explaining the forecast skill of every single case would however require more dedicated studies that are beyond the scope of this work.

**Discussion**

>> Discussion should be more developed addressing the existing capability of the weather forecast and so the ocean forecast to predict MHW (when generated by large-scale

atmospheric modes). Discuss the scales that can be addressed, both temporal and spatial. What are the limitations? What are the next steps and future? Expectations are very important due to the environmental and socio-economic implications, provide clear and more detailed skills and limitations, how is the ability of current forecast and which improvement can be expected? What about the incoming El-Nino and its impacts? The years 2022 and 2023 have reached records… Is the seasonal forecast with its ability to predict well ocean response during such events can help society for the coming months?

The discussion has been developed according to your comments, addressing the current strengths and weaknesses of seasonal forecast systems in predicting MHW in terms of representation of climate modes and ability to correctly capture the atmospheric and oceanic processes that impact the MHW occurrence at various spatial and temporal scales. A short discussion on how the recent surge in number and duration of MHW events and potential improvements of prediction systems can have huge socio-economic implications has been included in the Discussion section. See Section 4 of the manuscript.

**Reply to RC2**

>> The present manuscript (ms) evaluates the skills of the ECMWF SEAS5 system in predicting the MHWs. I find this study relevant for the journal scope, for the scientific community and mainly for the society and stakeholders. This analysis highlights the high capability to forecast the occurrence of the MHWs in some basins, although the authors also underlines how this study fails in some other regions.

>> The manuscript is well written, the analysis of the data is comprehensive, and results are promising. I recommend the publication after some minor revisions are taken into account.

We would like to thank you for taking the time to review our manuscript. We took into account your constructive comments and modified the manuscript accordingly. See below the responses to the individual comments.

>> Marine heatwaves (in the abstract, L21 and throughout the ms)

The manuscript was corrected accordingly.

>> L39: the work of Jacox et al 2022 shows how the forecast skill is regional dependent. How does it correlate with your work? Since the forecast skills described in this ms also varies regionally, are there similarity and/or differences? Pleas discuss it in the discussion section

Our results agree with Jacox et al (2022) in terms of regional skill. Large skill is found in the Tropics and the north-eastern Pacific, even at long lead time. Skill is much lower in the extra-Tropics and the Mediterranean. The main differences reside in the datasets, the methods and the MHW statistics. Jacox et al (2022) is more focused on the limits of the forecast range, while our manuscript aims at providing a simple way of providing a seasonal MHW forecast product that could be used operationally. This is now discussed in the manuscript.

>> L39: ENSO is not specified. You specify it later, in Line 41

ENSO was replaced by the eastern Equatorial Pacific.

>> L63: Why did you use the forecast starting days as Feb1st, May 1st , August 1st and November 1st? Please motivate it

We used these starting dates so that we cover seasonal predictions of Marine heatwaves in every season (MAM, JJA, SON and DJF) equally. This choice is now justified in the manuscript.

>>L69: ESA-CCI SST and not ESCA

The manuscript was corrected accordingly.

>> L79,84: why did you use 1996-2016 period? There is inconsistency with what is stated early in the manuscript (period of study  1982-2021)

We agree that the 1993-2016 reference period is different from the period of study (1982-2021). On the other hand, it is the same reference period used for the online C3S seasonal forecast charts that display the C3S multi-model seasonal forecast fields and verification scores (https://climate.copernicus.eu/charts/packages/c3s_seasonal/). This reference period is also commonly used for Copernicus Marine Service products as it covers the satellite altimetry observing period. It is also the period used for the recent study by McAdam et al (2023) on Marine heatwave seasonal forecasts in the north-eastern Pacific and the Mediterranean. This choice is now justified in the manuscript.

>> L92, 99: 1982-2020? Should be 2021?

Yes, the manuscript was corrected accordingly.

>> L99: …anomaly, and N is the total…

The manuscript was corrected accordingly.

>> L120, 131, 217: again, inconsistency with the period used

This was changed to 1982-2021.

>> Fig1: please specify in the caption what is JJA, MAM etc…

It is now specified in the caption as well as in Section 2.1.

>> The authors emphasize and show, that the forecast skills are very good in regions such El-Nino and Carribean, while in areas like the West Med and the North Sea, skills are poor. Can you please provide a better explanation for this? What can be the possible causes? These areas (e.g. El-Nino) are subject to high SST increase, but how about the atmospheric forcing? Are they considered? Probably in the poor skilled regions atmospheric forces are the driving mechanisms rather than the ocean conditions? Or, the definition of MHWs used is inconsistent? Everyone follow Hobday 2016, but probably a "regional" definition id needed? Have the authors thought about this? If fully skilled, this study would be very useful for the society and it will have positive impacts for the economy therefore, a more comprehensive explanation is needed.  Please, provide comments in the discussion section

Both Sections 3 and 4 have been reworked to address these points. The manuscript now discusses the impact of climate modes and both atmospheric and ocean processes working at various spatio-temporal scales on the occurrence and predictability of MHW in the locations we chose to focus on. Explaining the forecast skill of every single case would however require more dedicated studies that are beyond the scope of this work. Given the nature of this paper, we use a very generic definition of MHW, just based on objective temporal statistics. But different users may need different definitions or thresholds according to the location and/or the specific activity they are interested in. This aspect is now discussed at the end of the manuscript.

**List of all relevant changes in the revised manuscript**

In Section 2.1, more details on the ECMWF seasonal forecast system components and the SST forecasts (frequency, seasons) have been added.

In Section 2.3, the choice of both the reference period and the period of study is justified.

In Section 2.4, the ROC score is introduced with references to relevant publications.

In Section 3.1, Figures 1 and 2 are swapped and hence their respective descriptions.

Both Sections 3.1 and 3.2 now present the results in link with the relevant climate modes, climate changes and atmospheric and oceanic processes that may impact the occurrence and predictability of MHW.

Section 4 has been restructured including more discussion of the climate modes and processes involved in the occurrence and predictability of MHW. The section now includes comparisons to publications on MHW predictability and references to relevant publications. Potential improvements and avenues to extract more skill for MHW prediction are discussed in the last paragraph.

References to the datasets used in that studies have been added at the end of the manuscript (L.345-351).